# Strategies for primary HPV test-based cervical cancer screening programme in resource-limited settings in India: Results from a quasi-experimental pragmatic implementation trial

Anu Mary Oommen[1☯], Rita Isaac[2☯]*, Biswajit Paul[2], David Weller[3], Madelon L. Finkel[4], Anitha Thomas[5], Thomas Samuel Ram[6], Prashanth H. R.[2], Anne George Cherian[1], Vinotha Thomas[5], Vathsala Sadan[7], Rajeswari Siva[7], Anuradha Rose[1], Tobey Ann Marcus[1], Shalini Jeyapaul[1], Sangeetha Rathnam K.[2], Tabeetha Malini[1], Surenthiran N.[2], Paul Jebaraj[2], Neenu Oliver John[6], Charles Ramesh[2], Jeffers Jayachandra Raj C.[2], Rakesh Kumar S.[2], Balaji B. V.[2], Irene Dorathy P.[7], Valliammal Murali[2], Prema N.[7], Kavitha K.[1], Priya Ranjani D.[1]

1 Community Health Department, Christian Medical College Vellore, Tamil Nadu, India, 2 RUHSA Department, Christian Medical College Vellore, Tamil Nadu, India, 3 University of Edinburgh, Edinburgh, United Kingdom, 4 Weill Cornell Medical College, New York, New York, United States of America, 5 Department of Gynaecologic Oncology, Christian Medical College Vellore, Tamil Nadu, India, 6 Department of Radiation Oncology, Christian Medical College Vellore, Tamil Nadu, India, 7 College of Nursing Community Health, Christian Medical College Vellore, Tamil Nadu, India

☯ These authors contributed equally to this work.
* rita.isaac2013@gmail.com

**Data Availability Statement:** All relevant data are within the manuscript and its Supporting information files.

## Abstract

### Background

In order for low and middle income countries (LMIC) to transition to Human Papilloma Virus (HPV) test based cervical cancer screening, a greater understanding of how to implement these evidence based interventions (EBI) among vulnerable populations is needed. This paper documents outcomes of an implementation research on HPV screening among women from tribal, rural, urban slum settings in India.

### Methods

A mixed-method, pragmatic, quasi-experimental trial design was used. HPV screening on self-collected cervical samples was offered to women aged 30–60 years. Implementation strategies were 1) Assessment of contextual factors using both qualitative and quantitative methods like key informant interviews (KII), focus group discussions (FGDs), pre-post population sample surveys, capacity assessment of participating departments 2) enhancing provider capacity through training workshops, access to HPV testing facility, colposcopy, thermal ablation/cryotherapy at the primary health care centers 3) community engagement, counselling for self-sampling and triage process by frontline health care workers (HCWs). Outcomes were assessed using the RE-AIM (Reach, Effectiveness, adoption, implementation, maintenance) framework.

**Funding:** The research received financial support from the Prevent Cancer Foundation, US, 16/136/109, through 2021 global grants program. The funding agency played no role at any point in the research or publication process. The principal investigator and the corresponding author, Dr. Rita Isaac received the grant. The funder had no role in study design, data collection and analysis, decision to publish, or preparation of the manuscript. Web site of Prevent cancer foundation - www.preventcancer.org.

**Competing interests:** The authors have declared that no competing interests exist.

## Results

Screening rate in 8 months' of study was 31.0%, 26.7%, 32.9%, prevalence of oncogenic HPV was 12.1%, 3.1%, 5.5%, compliance to triage was 53.6%, 45.5%, 84.6% in tribal, urban slum, rural sites respectively. Pre-cancer among triage compliant HPV positive women was 13.6% in tribal, 4% in rural and 0% among urban slum women. Unique challenges faced in the tribal setting led to programme adaptations like increasing honoraria of community health workers for late-evening work and recalling HPV positive women for colposcopy by nurses, thermal ablation by gynaecologist at the outreach camp site.

## Conclusions

Self-collection of samples combined with HCW led community engagement activities, flexible triage processes and strengthening of health system showed an acceptable screening rate and better compliance to triage, highlighting the importance of identifying the barriers and developing strategies suitable for the setting.

## Trial registration

CTRI/2021/09/036130.

## Introduction

Cervical cancer is a disease of poverty, with 84% cases and 88% deaths occurring in low resource settings around the world [1]. Although age standardised incidence rates for cervical cancer have been decreasing since 1990, India has one of the highest rates, accounting for one-quarter of the worldwide burden of cervical cancer [2]. Screening at least 70% women aged 30–49 years twice, using a high performance test, 90% coverage with HPV vaccination and ensuring 90% treatment coverage for precancer/cancer are the strategies recommended by the World Health Organization (WHO) to eliminate cervical cancer as a public health problem by 2030 [3]. WHO guidelines now recommend transitioning to HPV testing followed by appropriate treatment, compared to the earlier cytology and visual inspection with acetic acid (VIA) approaches.

Despite the existence of VIA based cervical screening in the national program since 2010, for women aged 30–64 years, only 1.9% of Indian women aged 30–49 years reported having ever been screened for cervical cancer screening in 2019–2021 [4]. There is a pressing need for implementation research examining alternate feasible HPV-based screening strategies, community factors, and implementation challenges, especially in low resource settings in Low-and middle-income countries (LMICs). Considering the organisational and patient-related psychological and structural barriers that have contributed to poor VIA-based screening [5–7], self-collected HPV based approaches might prove to be the way forward for LMICs, including India, to improve screening coverage.

Implementation of HPV-testing in population-based screening programs will, however, be challenging–requiring new HPV testing systems, a sensitized community who will provide self-collected samples, management algorithms and referral systems to ensure appropriate treatment. This implementation research was carried out in low resource settings in Tamil Nadu, an Indian state with the second highest Disability Adjusted Life Year (DALY) rates for cervical cancer [2]. The target population included tribal, rural and urban slum women from 2

districts-Vellore and Tiruvannamalai. The objective was to identify the barriers and challenges faced by the providers and vulnerable populations in low resource settings and adopt strategies to implement a feasible HPV test based cervical cancer screening program.

## Methods

### Study setting

The study was carried out in 3 resource limited settings-rural, urban slum and tribal. Community health programs of the medical college where the study was conducted have been providing primary and secondary care services in the three study sites, including an organised VIA test based screening program in the rural site since 2006 [6], and sporadic efforts for cervical cancer screening in the urban slum and tribal sites. VIA based cervical cancer screening services have been provided through government run primary health care centres as well in these sites for more than a decade [8].

### Study design and population

This implementation science research used a pragmatic, quasi-experimental, pre-post design to evaluate the HPV test based cervical cancer screening implementation and program outcomes in three different low resource settings: rural (K.V. Kuppam rural block, Vellore), urban slum (Vellore city), and tribal areas (Jawadhi hills in Vellore and Tiruvannamalai districts).

Key implementation evaluation questions were the following:

1. Equitable coverage- How to reach hard-to-reach women in 3 resource limited settings?

2. What is the influence of cultural and access to health services' barriers on the individual and community-level acceptability of self-sampling HPV test based cervical cancer screen and treat programme?

3. Treatment Coverage—What percentage of screened-positive women complete timely follow-up? How much does the introduction of thermal-ablation and cryotherapy at the primary level increase women's access to and utilization of treatment?

Data were gathered, using a mixed methods approach, on the community perspectives, organisational capacity, screening acceptability, uptake, treatment coverage, and feasibility of the intervention.

In each site, villages or urban wards were chosen purposely with an enumerated population of 1000 to 1500 women aged 30–60 years, to be able to identify about 400 women per site for screening. In the rural site, eight villages were chosen with a population of 1437 women (30–60 years) who resided within 6 km of a rural secondary care center where follow up services were offered by a gynaecologist for HPV positive cases. From the urban slum, 20 streets, with a population of 1335 women, were chosen from two urban wards, where follow-up of HPV positive cases was offered in another rural secondary care center that was 5 km away. In the tribal hills, two sets of villages were selected: 11 villages within 10 km of a tribal primary care facility, where a gynaecologist provided follow up services (colposcopy and thermal ablation) once a month and five distant villages, about 20 km from the tribal primary care facility with a population of 1089 women aged 30–60 years Table 2.

The frontline healthcare workers (HCWs) chosen for this research were six newly recruited community health workers (CHWs) for the 16 tribal villages, 4 existing public health nurses for the 20 urban slums, and 4 existing field workers for the 8 rural villages. The capacity building program included the following steps: 1) Stakeholder engagement for capacity

development 2) Conducted training needs assessment 3) Implemented a capacity development response that included a HPV testing system (careHPV Test Kit, Qiagen) [9] was made available at the secondary care centre with a trained laboratory technician to process the test and conducted a series of 7 training workshops for the research team on HPV test self-sampling, one-on-one counselling, community education methods.

The study was approved by the Institutional Review Board and Ethics Committee of the medical college (IRB min. no. 14145, dated 1/07/2021), and all research was conducted according to the ethical principles of human research, Declaration of Helsinki. Informed written consent was obtained for the survey and procedures performed on women. The study was registered with the Clinical Trials Registry of India.

The screening program strategy included six essential components of a cervical screening program [10].

1. Identification of eligible women aged 30–60 years, with exclusion criteria of pregnancy, hysterectomy, and prior cervical cancer, using census data maintained by the health information systems of the community health programs.

2. Community engagement: Community meetings were arranged in each site, approximately two per week. Women, men, and adolescents were invited to these meetings, in which a preferred 20-minute video was played, explaining the anatomy of the female reproductive tract, diseases of the female reproductive system, symptoms of cervical cancer, and screening tests and treatment for cervical cancer. This was supplemented one-on-one counselling by public health nurses or CHWs on how to self-collect cervical samples, with pictorial charts, and demonstration of cervicovaginal self-sampling careBrush (Qiagen) and transport media. Other educational tools included posters and a one-page 'Option Grid' [11] with structured responses to frequently-asked-questions by the community to aid HCWs to counsel people for screening.

3. Maximising uptake: Self-collected cervico-vaginal sampling was used as the method of collecting samples. We facilitated consenting women to collect samples immediately after the group health education programs and others either at home or at the clinic visit later. Women in the tribal area often could not access group education programs due to scattered homes, geographic distance from the venue of educational campaigns, agricultural work till late evening and CHWs often not able to organise education programs in the late evening hours. Therefore, a home based education was offered to a small proportion of non-attendees in the tribal area to ensure equal opportunities for the most vulnerable.

4. Program operation was overseen by a central manager who coordinated training of research team, planned community education program schedules, facilitated transport of samples to secondary care center for HPV testing, and sent reports to each site for triaging. HPV testing could be done in the secondary care center using low cost Qiagen India Care HPV™ kit- a hybrid capture 2 test.

5. Follow up and treatment: CHWs /public health nurses/field workers informed the results to HPV positive women through phone calls and home visits, and referred them to the nearest centre, which provided colposcopy triage and thermal ablation/ cryotherapy. While women from rural and urban slum areas could report to the follow up centre on any day, tribal women were referred to once a month gynaecology clinic in the tribal health centre. The WHO follow-up algorithms that included colposcopy triage by nurses was used for premenopausal women (algorithm 6), and cytology as triage (algorithm 7) for post-menopausal women who were HPV positive [12]. Confirmatory colposcopy was performed

using the modified Swede's score, [13] to assess the need for ablative treatment by gynaecologists, except in post-menopausal women for whom cytology was used as a triage test. Treatment for triage positive women was thermal ablation in the clinics for the urban slum and tribal women, and cryotherapy in the rural clinic. For research purpose a biopsy was also done for those who gave consent. HPV negative women were informed of their results by phone or in person and were asked to be tested again after five years. Women in the tribal areas were informed of their results by CHWs only through home visits.

Evaluation of the program was done using the RE-AIM framework [14]. We assessed the reach of the program by screening coverage; effectiveness by the primary outcomes like proportion and variability across the three different settings of screen positive women who accessed triage tests and appropriate treatment; adoption of protocols by health staff and communities, as well as implementation adaptations made during program delivery in each setting were also assessed. Additional data from qualitative interviews with program managers and the checklist of standards for reporting implementation studies (StaRI) are given in the supplementary files.

Data was analysed using SPSS v. 24 (IBM). Descriptive statistics was used to describe categorical variables in frequencies, percentages and proportions with 95% confidence interval (CI), and continuous variables in mean and Standard Deviation (SD) or median and Interquartile Range (IQR). Comparative statistics used chi square test for proportions and t-test for normally distributed continuous variables and Mann–Whitney U test for non-normally distributed continuous variables between sub-groups. A p value of $< 0.05$ was considered statistically significant.

## Results

We developed an implementation research logic model (IRLM) with key research directions, determinants, implementation strategies, mechanisms, outcomes for advancing cervical cancer screening programme for LMICs "Table 1".

**Table 1. Implementation research logic model (IRLM): A causal model for effective screening program.**

| Determinants | Implementation strategies | Mechanisms | Outcomes |
|---|---|---|---|
| Community and participant perspectives, cultural beliefs, health seeking behaviour | Qualitative research | Key informant interviews Focus group discussions, KAP quantitative survey | Gained understanding of community perspectives and barriers on screening |
| Provider awareness, attitude and organizational capacity | Capacity building of providers and organization | Series of training workshops and online reviews, care HPV system procured and installed | Trained research team for screening; HPV testing colposcopy and treatment of preinvasive lesions facility in the centres; |
| Access to screening and prompt treatment | Triage and patient navigation; Colposcopy and thermal ablation/ cryotherapy facility at the community health canters and makeshift clinics closer to their homes | Colposcopy done by trained nurses at the outreach centres; Doctors provided cryotherapy/ thermal ablation/LEEP treatment | Around 30% of the women accessed screening in 8 months' time; compliance to colposcopy, thermal ablation/ cryotherapy ranged between 56 to 86.6% |
| Inconsistent access to health education | Health worker delivered community education | Video shows for structured teaching twice a week in all 3 settings | Increased access to information; improved awareness * |
| Availability of HPV testing and reporting | Self-sampling Use of Low cost careHPV (Qiagen) kit -hybrid capture test | HPV testing at the community health centre by existing lab technician | Facility to test HPV in the community health centre; HPV reporting system |
| Commitment of administrators | Needs assessment, dissemination workshop | Discussions and Sensitization talks | Successful completion of the project |

*Not included in this paper.

## The implementation of screening programme

During the eight month study period between December 2021 to July 2022, 1170 women aged 30–60 years made an informed choice to get screened.

**Descriptive characteristics of the participants.** The distance of communities from the health centre offering follow-up services, varied from 20 km for a distant tribal area to within 6 km. The women from tribal and urban slum sites reported a higher numbers of pregnancies compared to rural women and lower rate of cervical cancer screening in the last three years. Women from tribal site reported higher rates of gynaecological symptoms such as vaginal discharge and low abdominal pain compared to women from the rural and urban sites. It was noted that more than 75% of women who have accessed screening in the tribal area have not received any school education compared to less than 15% in urban slum and rural sites "Table 2".

**Reach and Effectiveness of the screening programme.** The screening participation rate was 30.3% overall within eight months, with 31.0% in the tribal, 26.7% in the urban slums and 32.9% in the rural sites. The prevalence of HPV positivity was highest among tribal women (12.1%, 95% CI: 8.6%–15.6%), followed by rural (5.5%, 95% CI: 3.4%–7.6%) and urban slum women (3.1%, 95% CI: 1.3%–4.9%). The follow up rate for undergoing triage testing was higher in the rural site (84.6%), compared to tribal (56.1%) and urban slum sites (45.5%). The follow-up rate in the tribal area could be increased to 56% only with the special follow camps organised in the far away villages in a local school, taking colposcopy and thermal ablation to the doorsteps of the women. Of those who attended the triage visit, rate of colposcopy detected lesions was higher in tribal areas (17.4%,) compared to rural (9.1%), with no woman from the urban site detected to have lesions. Of the six biopsies done, three had high grade squamous intraepithelial lesions, two had low grade squamous intraepithelial lesions and one was negative.

Reported preference for self-collection as a mode of screening increased significantly in tribal areas from 11% to 30% (p = 0.001), with a smaller non-significant increase in rural areas (10% to18%, p = 0.153), and a non-significant decrease in urban slum (7% to 2%, p = 0.169). It was interesting to note that women in the rural and urban slum sites preferred healthcare professional collecting the samples if given an option. In all the three sites, the most preferred

**Table 2. Descriptive characteristics of the screened women participants.**

|  | Tribal site | Urban Slum site | Rural site |
|---|---|---|---|
| Distance from follow up testing centre | 1–20 km | 5–6 km | 2–6 km |
| Number of women accessed screening | 339 | 357 | 474 |
| Age (median, IQR) | 38 (33–45) | 39 (34–47) | 41 (35–48) |
| Mean (SD) pregnancies | 3.1 (1.50) | 3.1 (1.58) | 2.5 (1.02) |
| Education (N, %) |  |  |  |
| No education | 256 (75.5) | 74 (20.7) | 50 (10.5) |
| Primary school (1–5) | 27 (7.9) | 105 (29.4) | 74 (15.6) |
| Middle school (6–8) | 21 (6.2) | 71 (19.9) | 111 (23.4) |
| High School & Higher secondary (9–12) | 31 (9.1) | 86 (24.1) | 194 (40.9) |
| Graduate and Post-Graduate | 4 (1.2) | 21 (5.9) | 45 (9.5) |
| History of screening in the past 3 years | 11 (3.2) | 11 (3.1) | 78 (16.5) |
| History of Vaginal discharge | 57 (16.8) | 24 (6.7) | 17 (3.6) |
| History of Lower abdominal pain | 84 (24.8) | 18 (5.0) | 16 (3.4) |
| History of Dysuria | 37 (10.9) | 6 (1.7) | 49 (10.4) |
| History of Pruritis vulva | 20 (5.9) | 12 (3.4) | 35 (7.4) |

**Table 3. Implementation outcomes of the screening program.**

| Variables | Tribal | Urban Slum | Rural |
|---|---|---|---|
| Eligible population in the study site for screening | 1089 | 1335 | 1437 |
| Total screened (uptake rate) | 339 (31.0%) | 357 (26.7%) | 474 (32.9%) |
| HPV test positivity rate | 41 (12.1%) | 11 (3.1%) | 26 (5.5%) |
| Compliance to follow up of screen positives | 23 (56.1%) | 5 (45.5%) | 22 (84.6%) |
| Colposcopy positive among those triaged | 4 (17.4%) | 0(0%) | 2(9.1%) |
| Treated by thermal ablation/cryotherapy (rural site) | 2 | - | 1<br>1 |
| Conisation | 1 | - | |
| Treated by antibiotics | | | 1 |
| Refused treatment | 1 | - | - |
| Received treatment as by WHO protocol among triaged | 22/23 (95.7%) | 5/5 (100%) | 22/22 (100%) |
| Received appropriate treatment among all HPV positives | 22/41 (53.7%) | 5/11 (45.5%) | 22/26 (84.6%) |

modes of receiving health information were movies/TV or video shows, and one-on-one education and there was an increase in preference for the former, after the program. Other modes of health education such as reading handbills and receiving phone messages were preferred more by rural participants compared to tribal and urban poor "Tables 3 and 4".

**Adoption and implementation fidelity.** Of the six women (four tribal, two rural) who had colposcopy detected lesions, two had Swede scores 5 and 7. They received immediate thermal ablation; the woman who had a swede score of 7 refused to go to a tertiary care centre for LEEP (recommended according to the protocol). One woman underwent conization (Swede score 6); one received cryotherapy at the rural centre (Swede score 6), one who got a Swede score of 5 and had a low grade lesion on biopsy received only antibiotics going by the earlier protocol followed in the centre (poor adoption of program protocol), and one refused treatment.

The median time from receiving HPV report to follow-up assessment was 36 days (IQR: 18–51), with 28 days in tribal (IQR: 12–61), 36 days in urban slum (IQR: 31–44) and 36 days in rural areas (IQR: 14–50). Although the median days to follow up was similar in all areas,

**Table 4. Preferences of the community regarding screening and education.**

| Domain | Tribal | | Urban poor | | Rural | |
|---|---|---|---|---|---|---|
| | Pre (100) | Post (100) | Pre (100) | Post (100) | Pre (100) | Post (100) |
| Preference of who should do the screening (multiple responses) | | | | | | |
| Self-collection for HPV test | 11 | 30 | 7 | 2 | 10 | 18 |
| Nurses | 21 | 38 | 58 | 49 | 41 | 55 |
| Doctors | 61 | 42 | 45 | 55 | 69 | 73 |
| Either nurses or doctors | 20 | 18 | 32 | 23 | 10 | 15 |
| Health workers | 3 | 18 | 1 | 3 | 3 | 2 |
| Most preferred methods and tools for health information | | | | | | |
| One-on-one education | 55 | 34 | 50 | 41 | 32 | 47 |
| TV/ video shows/movies | 27 | 45 | 22 | 39 | 28 | 44 |
| Print mass media | 0 | 8 | 14 | 15 | 15 | 4 |
| Street play/puppet shows | 2 | 9 | 4 | 0 | 6 | 3 |
| Sharing of experiences by peers | 4 | 4 | 8 | 2 | 10 | 1 |
| Others | 1 | 0 | 6 | 3 | 5 | 1 |

some women in tribal areas had long delays (maximum delay of 152 days). The reasons for longer delays in the tribal area included hesitancy in accepting treatment despite repeated counselling, absence of any relative willing to bring the woman for follow up, temporary migration for work.

**Implementation challenges and solutions.** Patient-related and health system challenges faced in implementing the screening program are outlined in Table 5.

Attempts were made to overcome challenges, by discussing with local health workers and service providers how best to devise appropriate solutions. The tribal areas faced specific challenges due to work related temporary migration, once a month only gynaecology clinics at the primary care centre, poor access to public transport, and lack of social support. Special camps held in the distant tribal site at the makeshift clinic closer to their homes by the doctor-nurse

**Table 5. Programatic change and strategies adopted to address barriers to screening, triage and follow up (data from qualitative inquiry of program managers).**

| Barriers (with example quotes) | Solutions |
|---|---|
| **Identification and invitation of eligible women** | |
| Outdated census information | Updated census of each site to identify denominator |
| Difficulty in reaching eligible tribal women | Various sources used to contact women for gatherings |
| Women not available in the daytime *"evening is the best time to give mass health education"* | Evening sessions arranged for health education |
| **Screening test and laboratory issues** | |
| Some women worried about the brush *"whether it will cause injury or something"* | Counselling |
| **Follow-up challenges related to target population** | |
| Unwillingness (fear) of women to undergo internal (pelvic) examination when they have no symptoms | Repeated counselling/rapport building by nurses, social worker, health workers |
| Poor family support | Counselling family |
| Regular menstrual cycles coinciding with doctors' monthly clinic days at the primary care centre (*Specific to tribal*) | Provided transport costs for women to come to a secondary care centre where diagnostic facilities available daily |
| Migration of tribal women to another state for a few months at a time | Women are followed up after they have returned |
| Phone connectivity issues (tribal) *"because of lack of connectivity, even the women who had phones we could not call and tell even negative results; we could not tell them the result has come"* | Home visits to inform results—tribal |
| **Follow up challenges related to the health system** | |
| Protocol deviations during follow up visits (e.g., missed ablation after positive colposcopy) | Treatment done based on biopsy (specimen collected at the time of colposcopy) |
| Primary care clinic far away >10 km for people in 11 villages in the tribal site. Lack of public transport from these villages to clinic, poor roads | Special colposcopy camp by a trained nurse organised along with routine mobile clinics in makeshift clinics (school buildings) in the remote villages |
| Gynaecology clinic in tribal primary care facility only once a month | Gynaecologist made special visits to the villages that were more than 10 km away, for confirmatory colposcopy and ablation at the makeshift clinics held in a school building. Vehicle was sent to various points to pick up HPV positive women and transported them to the makeshift clinics. |
| Long waiting time for colposcopy in monthly gynaecology clinic (tribal) *"After waiting three hours alongside pregnant women these women left without assessment as they would miss the bus back."* | Staff directed to prioritise HPV positive women |

teams for colposcopy and thermal ablation and special transport arranged for clinic visits over-came some of these issues.

## Discussion

HPV testing has become the primary screening modality for most high income countries following the recommendation of WHOs essential practice guidelines in 2014 [15,16] and more so after the launch of the WHO strategy for eliminating cervical cancer as a public health problem in 2018 [17]. However, India and most LMICs are yet to adopt HPV test as the primary screening test for lack of adequate information on contextual factors that contribute to the adoption of these evidence based interventions from implementation research in different settings in LMICs [18]. In a recent commentary, Prajakta Adsul et al noted the WHO working group's recommendation on need for more implementation research with a specific focus on broadening stakeholder engagement, equity in reach and navigation throughout the continuum of care, and improving the readiness of health systems to increase adoption and implementation of cervical cancer screening programs. The authors highlighted the key research advances within the resource-limited settings that could contribute towards enhancing the science of implementation around global cervical cancer screening efforts [19,20].

We designed this study to answer the key implementation research questions for rolling out effective hrHPV self-sampling test based cervical cancer screening in 3 different resource limited settings in India. It was found that about 30% of women accessed screening in 8-months' time across rural, urban slum, tribal settings using the self-sampling HPV test which is quite a high uptake rate as HPV testing needs to be done only once in 5 years. Whereas the ongoing cervical cancer screening program in the rural site using visual inspection methods (VIA/VILI) for the last 10 years, has shown that annually, only about 6–8% of eligible women accessed screening [6]. The challenges of administering VIA based screening were the following: The rural community health centre covering a population of 130,000 could only organise a maximum of 24 screening camps in a year as part of their ongoing outreach health care services, with about 30 to 50 women participating in each camp, totalling 800–1200 women getting screened every year due to other health service commitments and shortage of personnel. It has taken about 10 years to screen 60% of the estimated 15,000 eligible women at least once [Data from the health information system of study organisation].

HPV test-self-sampling strategy addressed the shortage of trained personnel in administering visual inspection method based screening at the outreach centres, lack of suitable infrastructure for gynaecological examination, poor organizational capacity to organise multiple screening camps and embarrassment of women to undergo gynaecological examination. The high screening uptake rate in this study shows that self-sampling for HPV test was acceptable to women from 3 different resource limited settings in this region. Several feasibility studies done in the past in LMICs have provided substantial evidence for the effectiveness of self-sampling for HPV test based cervical cancer screening, [21–24] however there is a scarcity of published articles on the other contextual factors that contribute to the uptake and implementation of HPV self-collected sample based screening and compliance to triage and treatment.

### Community factors and programme adaptations that improved the screening uptake and compliance to triage

The programme used the preferred video shows with structured teaching programme as the tool for community awareness campaigns, organised in local school buildings and public halls

and that was delivered by the HCWs reducing the cost of community education and increasing the population coverage within a short period of time.

In public health care delivery system across the world, HCWs form an important link and resource between the community and the health care system. As most of the time, these HCWs live in the community they serve and, therefore, they are familiar with the social and cultural practices of members in the community. HCWs form an effective frontline health workforce for the medically underserved populations [25–27]. The HCWs were able to complete 191 educational sessions over 8 months with a total of 58 men and 2391 women participating in these campaigns. The education tools also included an 'Option Grid' with structured responses to frequently asked questions by the community regarding the cervical cancer screening programme, for the use of HCWs [11].

## Flexibility in service delivery and patient navigation

We found that a deeper understanding of disparities in screening uptake across 3 resource limited settings and a move towards equity orientation will make the screening programme more acceptable and successful. Towards addressing the disparities, we offered colposcopy and treatment (thermal ablation) at the makeshift clinics closer to their homes in the tribal area. We identified the assets in the community like the health care workers to be patient navigators, promoted a collaborative and equitable partnership with multidisciplinary team members in all 3 participating departments in all phases of research and fostered co-learning and capacity building.

Our study had a few limitations, such as restriction to a single state in south India, lack of a control arm, non-involvement of government primary health centres for program delivery and non-assessment of cost effectiveness analysis. Though a quasi-experimental design without the true control groups has limited scope to conclude a causal association between an intervention and an outcome, its high external validity, ability to answer the key implementation science research questions like feasibility, adoption, fidelity and acceptability of the intervention in real world settings, make the design most appropriate for this study. A quasi-experimental design was chosen as random assignment was difficult or rather impossible in field settings. Unlike a true experiment, a quasi-experiment does not rely on random assignment, instead, subjects were part of naturally existing clusters (rural, urban slum and tribal) in India, based on non-random criteria. We have chosen the 3 resource limited settings within the existing geographic divisions in India. Though there may be some sociocultural differences in the uptake of the interventions across states, the 3 study sites largely represent rural, tribal and urban slums, the major geographical divisions in India. The study did not report any harm and unintended effects due to the intervention.

## Conclusions and recommendations

This implementation science research examined and addressed the complexities associated with cervical cancer screening in 3 different resource-limited settings in India. We identified the contextual factors that contributed to the implementation challenges and feasible solutions to HPV screening in real life settings. We believe that findings from our study will be an invaluable resource to government (local, state, and federal), policy makers, and healthcare providers in India to inform policy decisions regarding adoption of HPV-based cervical cancer screening. The path to achieving population-level benefit from existing evidence-based interventions for effective cervical cancer screening, is clear only if we are to see it through an implementation research perspective. To achieve the goals of screening, we need to accelerate

the bidirectional exchange of information and share lessons learned as we gather a deeper understanding of context surrounding resource-limited settings.

## Supporting information

**S1 Checklist. Standards for reporting implementation studies: The StaRI checklist for completion.**
(DOCX)

**S1 File. Code book of qualitative data.**
(DOCX)

## Acknowledgments

The authors acknowledge the contribution of the community health workers who supported the screening program and all the participants in the survey who provided invaluable insights into how to implement the screening program.

## Author Contributions

**Conceptualization:** Rita Isaac, David Weller.

**Data curation:** Anu Mary Oommen, Rita Isaac, Biswajit Paul.

**Formal analysis:** Anu Mary Oommen, Rita Isaac, Biswajit Paul.

**Funding acquisition:** Rita Isaac, Madelon L. Finkel.

**Investigation:** Rita Isaac, Biswajit Paul, Prashanth H. R., Vinotha Thomas.

**Methodology:** Rita Isaac, Biswajit Paul, David Weller, Anitha Thomas, Thomas Samuel Ram, Vinotha Thomas.

**Project administration:** Anu Mary Oommen, Rita Isaac, Biswajit Paul, Prashanth H. R., Anne George Cherian, Vathsala Sadan, Rajeswari Siva, Anuradha Rose, Tobey Ann Marcus, Shalini Jeyapaul, Sangeetha Rathnam K., Tabeetha Malini, Surenthiran N., Paul Jebaraj, Charles Ramesh, Jeffers Jayachandra Raj C., Rakesh Kumar S., Balaji B. V., Irene Dorathy P., Prema N., Kavitha K., Priya Ranjani D.

**Resources:** Rita Isaac, Biswajit Paul, Madelon L. Finkel, Anitha Thomas.

**Software:** Rita Isaac.

**Supervision:** Anu Mary Oommen, Rita Isaac, Biswajit Paul, Anne George Cherian, Shalini Jeyapaul, Sangeetha Rathnam K., Surenthiran N., Paul Jebaraj, Neenu Oliver John, Charles Ramesh, Jeffers Jayachandra Raj C., Rakesh Kumar S., Balaji B. V., Irene Dorathy P., Valliammal Murali, Prema N., Kavitha K., Priya Ranjani D.

**Validation:** Anu Mary Oommen, Rita Isaac, Biswajit Paul.

**Visualization:** Rita Isaac.

**Writing – original draft:** Anu Mary Oommen.

**Writing – review & editing:** Rita Isaac, Biswajit Paul, David Weller, Madelon L. Finkel, Anitha Thomas, Thomas Samuel Ram, Prashanth H. R., Anne George Cherian, Vinotha Thomas, Vathsala Sadan, Rajeswari Siva, Anuradha Rose, Tobey Ann Marcus, Shalini Jeyapaul, Sangeetha Rathnam K., Tabeetha Malini, Surenthiran N., Paul Jebaraj, Neenu Oliver John,

Charles Ramesh, Jeffers Jayachandra Raj C., Rakesh Kumar S., Balaji B. V., Irene Dorathy
P., Valliammal Murali, Prema N., Kavitha K., Priya Ranjani D.

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
