## [Decision Letter · Decision Letter 0]

11 Jan 2024

PONE-D-23-25033Strategies for primary HPV test-based cervical cancer screening programme in resource-limited settings in India: results from a quasi-experimental pragmatic implementation trialPLOS ONE

Dear Dr. Isaac,

Thank you for submitting your manuscript to PLOS ONE. After careful consideration, we feel that it has merit but does not fully meet PLOS ONE’s publication criteria as it currently stands. Therefore, we invite you to submit aPlease ensure that your decision is justified on PLOS ONE’s publication criteria and not, for example, on novelty or perceived impact.

We look forward to receiving your revised manuscript.

Kind regards,

Pijush Kanti Khan, Ph.D.

Academic Editor

PLOS ONE

Reviewers' comments:

Reviewer's Responses to Questions

**Comments to the Author**

1. Is the manuscript technically sound, and do the data support the conclusions?

Reviewer #1: Yes

Reviewer #2: Yes

2. Has the statistical analysis been performed appropriately and rigorously? 

Reviewer #1: Yes

Reviewer #2: Yes

3. Have the authors made all data underlying the findings in their manuscript fully available?

Reviewer #1: Yes

Reviewer #2: Yes

4. Is the manuscript presented in an intelligible fashion and written in standard English?

Reviewer #1: Yes

Reviewer #2: Yes

5. Review Comments to the Author

Reviewer #1: First and foremost I would like to thank you for providing me an opportunity to review this research article which is a piece of outstanding work of implementation research (IR). I am very pleased to inform you that this study will bring an outstanding outcome in terms of screening coverage, treatment and compliance in low resource setting in tribal, rural and slum-urban areas of India which is a very neglected areas of our Indian society.

Barriers and issues were very well addressed and explained in context of tribal, rural and slum-urban settings highlighting the challenges in real settings in terms of Indian population.

Methodology is very clear and well explained to achieve the target of research question for evaluation of implementation programmes.

Framework of IR is well addressing the contextual factors and is found to be feasible, adaptable and scalable to limited resource settings of the present evidence based intervention.

Barriers and challenges are very well identified and addressed and programmatic changes were made to implementation strategy for its adoption in the real setting for optimization of HPV test based cervical cancer screening, and its treatment programmes.

A profound and robust implementation strategy were adopted to provide large coverage to women population of the study settings. Multi-faceted approach were adopted to increase the accessibility of HPV test, health care provider capacity building, community engagement, counselling of community by self sampling by the healthcare provider and triage process by frontline health care workers (HCWs).

After implementation outcome, a significant proportion of compliance to triage process was also reported in the study settings which is a signification solution of applying implementation strategy and translating research into practice.

Findings of the present study will contribute a significant outcome of HPV based cervical cancer screening and its treatment programme through IR ensuring its adoption, fidelity, feasibility, adaptability, scalability in terms of women population coverage to maintain the sustainability in the limited resource settings in other states of India as well.

Reviewer #2: Thanks for the invitation to review this study. This research evaluates the implementation of HPV test-based cervical cancer screening in rural, urban slum, and tribal areas in India. It employed a pragmatic, quasi-experimental design, focusing on self-collected HPV testing for women aged 30-60. The study found varied screening and triage compliance rates across settings, highlighting the importance of adapting strategies to local contexts. Key challenges included accessibility, cultural barriers, and health system limitations. The study underscores the feasibility and need for context-specific adaptations in cervical cancer screening programs in low-resource settings.

The study identifies several limitations. Firstly, its quasi-experimental design limits the ability to establish causality. Secondly, self-reported data may introduce recall bias and affect the reliability of some findings. Thirdly, the study focuses on specific geographic areas in India, which may limit the generalizability of the results to other settings. Additionally, the study does not fully explore the long-term outcomes of the screening program, and the potential impact of socio-cultural factors on screening behaviours is not thoroughly examined. Finally, the study's reliance on existing health infrastructure might overlook the challenges in settings with more limited resources. Adding these items to the limitations section of the manuscript would make it more informative for the audience. Other sections and aspects of the manuscript is well prepared and I have no further comments and suggestions for improvement.

6. PLOS authors have the option to publish the peer review history of their article (what does this mean?). If published, this will include your full peer review and any attached files.

Reviewer #1: **Yes: **Dr. Anjali Kumari

Reviewer #2: **Yes: **Sina Azadnajafabad, MD, MPH

---

## [Author Response · Author response to Decision Letter 0]

24 Feb 2024

Response to Reviewers 

On behalf of all the authors, I want to place on record our sincere thanks to the reviewers for an excellent review and appreciation of our hard work and study methods chosen. We see the reviewers have a good understanding of the implementation science research. We want to thank the editors too for identifying the right people for the review. 

Response to academic editor’s review 

• We haven’t made any changes in the financial disclosure. 

• We do not have any figure in the manuscript. 

• We have used publicly available, routinely used laboratory tests for this research, so depositing the laboratory protocols in protocol.io is not applicable. 

• We have followed the PLOS ONE's style requirements while formatting to the best of our ability

• We have removed the reference no.21 from the previous submitted version as it is not a published article. It is a reference to data from the existing health information system of the research organisation. We have mentioned that in parenthesis in the text. 

 21. Health information system, RUHSA. Christian Medical College Vellore. 

• We have edited the references 4 and 5 to comply with PLOS ONE publication criteria 

Reviewer’s comments

Reviewer 1#

We would like to thank the reviewer once again for the excellent review and appreciation of our work and study methods. 

Reviewer 2#

1. The study identifies several limitations. Firstly, its quasi-experimental design limits the ability to establish causality.

As the reviewer has suggested, we have added a few additional lines to explain why a quasi-experimental design was chosen and its merits in 342 to 355 lines.

2. Secondly, self-reported data may introduce recall bias and affect the reliability of some findings.

We have mentioned in the methodology section that the study focused on rigorous training of the data collectors to reduce the recall bias. 

Also the intervention was rolled out only for a short period of 8 months and therefore pre-post intervention qualitative data collected on the barriers and challenges of cancer screening may not be significantly affected by recall bias. 

As the main outcomes we measured were screening and follow-up rates which were objective measures based on our program data, these are not affected by recall bias. 

3. Thirdly, the study focuses on specific geographic areas in India, which may limit the generalizability of the results to other settings.

Across all states in India, the society has been broadly divided into tribal, rural and urban societies on the basis of their geographical surroundings and socio-cultural characteristics. We have chosen the 3 resource limited settings within the existing geographic divisions. Though there may be some sociocultural differences in the uptake of interventions across states, the 3 sites largely represent rural, tribal and urban slums, the major geographical divisions in India 

4. Additionally, the study does not fully explore the long-term outcomes of the screening program, 

This study is a short term pragmatic pilot trial. There is a fund approved RCT being planned in the study site which will also measure long term outcomes. 

5. the potential impact of socio-cultural factors on screening behaviours is not thoroughly examined. 

We believe the differences that this study have picked up in 3 diverse societies explain the impact of sociocultural factors. We have also examined these through qualitative studies in the area before and after the intervention, which will be published separately.

6. Finally, the study's reliance on existing health infrastructure might overlook the challenges in settings with more limited resources. 

We agree with the reviewer that the existing health infrastructure in the study sites would have certainly impacted the outcomes. It will be more challenging to implement this screening programme in more limited settings. 

7. Adding these items to the limitations section of the manuscript would make it more informative for the audience.

As per the reviewer’s suggestion, we have added a few lines under limitations. Please see the lines 342 to 355. 

Thank you,

Prof. Rita Isaac

Corresponding Author

---

## [Editor Report · Decision Letter 1]

14 Mar 2024

Strategies for primary HPV test-based cervical cancer screening programme in resource-limited settings in India: results from a quasi-experimental pragmatic implementation trial

PONE-D-23-25033R1

Dear Dr. Isaac,

We’re pleased to inform you that your manuscript has been judged scientifically suitable for publication and will be formally accepted for publication once it meets all outstanding technical requirements.

Kind regards,

Pijush Kanti Khan, Ph.D.

Academic Editor

PLOS ONE
---

## [Editor Report · Acceptance letter]

26 Mar 2024

PONE-D-23-25033R1 

PLOS ONE

Dear Dr. Isaac, 

I'm pleased to inform you that your manuscript has been deemed suitable for publication in PLOS ONE. Congratulations! Your manuscript is now being handed over to our production team.

Kind regards, 

on behalf of

Dr. Pijush Kanti Khan 

Academic Editor

PLOS ONE